

# Aerosol optical properties as observed from an ultralight aircraft over the Strait of Gibraltar

Patrick Chazette

[1]Laboratoire des Sciences du Climat et de l'Environnement (LSCE), Laboratoire mixte CEA-CNRS-UVSQ, UMR 1572, CEA Saclay, 91191 Gif-sur-Yvette, France

*Correspondence to*: Patrick Chazette (patrick.chazette@lsce.ipsl.fr)

**Abstract.** An unprecedented scientific flight was conducted over the Strait of Gibraltar to study the optical properties of the atmospheric aerosols from the sea surface to the lower free troposphere within the framework of the southern Spain experiment for spaceborne mission validation (SUSIE). A Rayleigh-Mie lidar was installed on an ultra-light aircraft (ULA) for vertical (nadir) and horizontal line-of-sight measurements. This experiment took place on 13 August in parallel with continuous observations with a $N_2$-Raman lidar from the coastal site of San Pedro Alcantara (~50 km north-east of Gibraltar). Significant differences were observed between the optical properties of the aerosol layers sampled over the Strait of Gibraltar and San Pedro Alcantara. These differences are related to the surface/atmosphere interface in the planetary boundary layer and as well as the origins and transport processes in the lower free troposphere. A significant contribution of terrigenous aerosols originating from the Iberian Peninsula is highlighted over the two areas. These polluted dusts are identified with lidar ratios (LR) ~45±8 sr higher than those of Saharan aerosols sampled during the same period (<34 sr) at 355 nm. Furthermore, the particle depolarization ratio is derived with values of ~10-15% for the polluted dust and >20% for the Saharan dusts. The difference in LR is the opposite of what is usually assumed for these two types of aerosols and highlights the need to update the classification of aerosols in terms of LR to be used in the inversion of vertical profiles from future spaceborne missions embedding a lidar operating at 355 nm.

**Keywords:** Aerosols, Gibraltar, ultralight aircraft, lidar, optical properties, lidar ratio, depolarization ratio

## 1 Introduction

Very little data exist on the aerosol characterization above the Strait of Gibraltar and its surroundings area where the Atlantic Ocean and the Mediterranean Sea meet. The temperature difference between these two-water surfaces inevitably induces a specific atmospheric circulation within the lower troposphere associated with the well-known low-pressure corridor from the Atlantic Ocean to the Mediterranean Sea. The vertical distribution of aerosols can therefore be very heterogeneous against time and space in this region, and is an exciting source of study. In addition to this, the Mediterranean region is identified as one of the "hotspots" in projections of future climate change (Giorgi and Lionello, 2008) where aerosol direct and semi-direct effects are not properly taken into account in global climate change simulations (IPCC, 2014). Indeed, the presence of aerosols in the lower and middle troposphere have a significant impact on sea surface temperature, evaporation and precipitation at the regional scale (e.g. Nabat et al., 2015). This impact is mainly felt through a probable positive feedback on the trend for future dryer and thus more atmospherically turbid Mediterranean summers.

In order to characterize the vertical distribution of aerosols over a long period, from mid-June to the end of August 2011, a ground-based remote sensing station was therefore installed in southern Spain, in the municipality of San Pedro Alcantara (36°29'11" N 4°59'33" W), near Marbella, in Andalusia. This installation was one of the components of the FENNEC program which was conducted from June to July 2011 (Ryder et al., 2013) and was specifically extended by



the southern Spain experiment for spaceborne mission validation (SUSIE) to support an airborne experiment planned on August 2011 over the Strait of Gibraltar. This airborne experiment was funded by the Centre National d'Etude Spatiales (CNES) and the Commissariat à l'Energie Atomique et aux Energies Alternatives (CEA). Its main goal was the preparation of the validation campaign for 355-nm wavelength Earth observation space missions such as the Atmospheric Dynamics Mission Aeolus (ADM-AEOLUS), which was launched in August 2018 (Stoffelen et al., 2005; Andersson et al., 2008). It is also a powerful tool towards the preparation of future Earth Clouds, Aerosols and Radiation Explorer mission (EarthCARE, Illingworth et al., 2015), which is part of the Living Planet program of the European Space Agency (ESA) and for which upstream studies based on simulations (e.g. Chazette et al., 2001, 1998) have been conducted in the past to size the ATmospheric LIDar (ATLID) planned to be embedded onboard the EarthCARE satellite. Therefore, it appeared necessary to rely on actual observations of different scientifically relevant atmospheric environments to build a robust database of vertical lidar profiles at 355 nm in order to conduct furthermore realistic link budget studies. The SUSIE experiment is part of this objective and preceded the Chemistry-Aerosol Mediterranean Experiment (ChArMEx, http://charmex.lsce.ipsl.fr) which took place in the western Mediterranean in 2013-2014 (Mallet et al., 2015) over the Balearic Islands (Ancellet et al., 2016; Chazette et al., 2016), Lampedusa island (e.g. Meloni et al., 2018) and the French Riviera (Chazette et al., 2019). It differs from ChArMEx by its location which is at the far west of the Mediterranean Sea in connection with the Atlantic Ocean.

The choice of the Strait of Gibraltar and Andalusia to conduct this field campaign was dictated by the high variability in optical thicknesses and aerosol type that can be encountered over this geographical area, as shown by Rodríguez et al (2001) via ground measurements. This variability is closely linked to the diversity of sources, but also to highly variable meteorological conditions over time (Gallero et al., 2006). The objective of this paper is to present the original results obtained from this field experiment. It brings a piece of information towards the understanding of the variability in the lower troposphere of both the optical properties and origins of observed aerosols. We will see that this is not as predictable as one might expect.

In section 2, we present the instrumental configuration. The methods used to derive the optical properties of aerosols from the lidar profiles are explained in section 3. The analysis of the observations is carried out in section 4 and section 5 presents the origin of the aerosols observed over Gibraltar during the airborne experiment. In Section 6, we summarize and conclude.

## 2    Instrumental set up and strategy

Two observational platforms are used in this work, one is airborne and the other one is positioned at the ground level. Hereafter, we present the instruments which compose those platforms.

### 2.1    Airborne measurements

Airborne measurements over the Strait of Gibraltar were performed using an Ultra-light aircraft (ULA) equipped with an active remote sensing device. In this paragraph, the ULA scientific payload is presented as the flight plane used during the experiment.

### 2.1.1    Payload

The ULA/Tanarg-installed eye-safe lidar system ULICE (Ultraviolet LiDAR for Canopy Experiment) (Shang et al., 2016) was used  to study the lower troposphere (between ~0.15 and 3 km above the mean sea level (AMSL)) over the Strait of Gibraltar. The lidar and the ULA's flights close to Gibraltar are represented in Figure 1. The Tanarg 912 XS was built by the Company Air Création (http://www.aircreation.fr/) and offers a maximum total payload of ~250 kg including a scientific payload of ~120 kg with a maximum autonomy of ~3 hours.  The cruise speed is around 85-90 km h$^{-1}$ and the



maximum flight altitude is ~6 km AMSL (Chazette et al., 2007). The ULA is also equipped with a Global Positioning System (GPS) and an Attitude and Heading Reference System (AHRS), which are part of the MTi-G components from XSens (https://www.xsens.com/).

The data acquisition was performed by averaging 1000 laser shots at 100 Hz pulse repetition frequency, leading to a
temporal sampling close to 10 s. The lidar is controlled by a custom Labview software on a PCI eXtensions for Instrumentation (PXI) computer (National Instruments, http://www.ni.com). The ULA payload is autonomous with a power supplies by an alternator associated with the propeller. It can deliver the required 600-700 W.

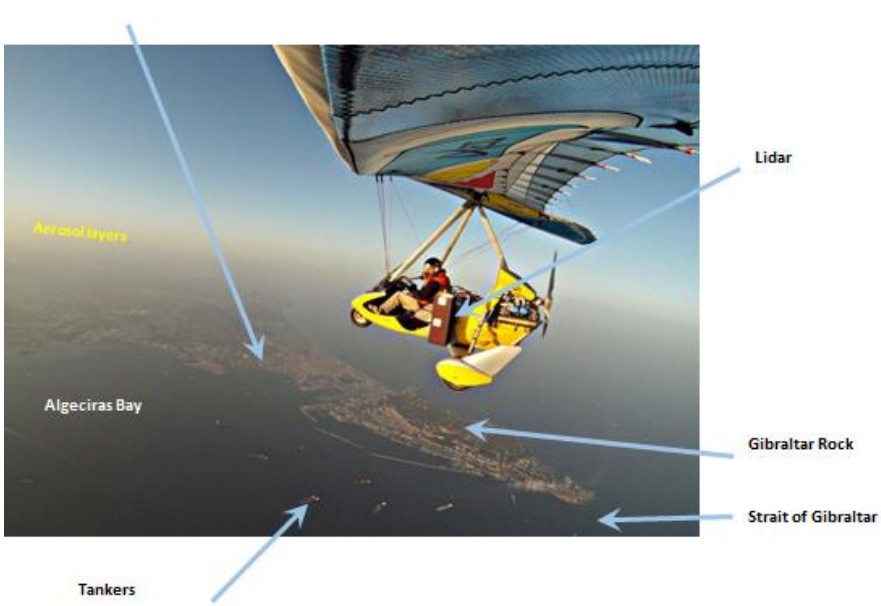

**Figure 1: ULA above the Strait of Gibraltar. The lidar is located on the left side of the ULA in nadir shooting position. The**
**aerosol layer can be seen in ochre on the horizon of the photo.**

### 2.1.2 Flight plan

The flight plan allowed a maximum number of lidar profiles during the fllight over the Strait of Gibraltar. There were 2 phases during the flight, a first phase where the lidar line of sight was horizontal and a second phase where the lidar shots were made with nadir sighting (Figure 2). It was during the ascent and descent that the lidar's line of sight was horizontal
to allow the vertical profile of the aerosol extinction coefficient to be reconstructed without assumptions about aerosol types, as shown by Chazette et al. (2007). After take-off from a private field aerodrome (Tahavilla, Spain, 36°11' 17.8" N 5°45'23.2" W), the ceiling for the flight was reached between 2.5 and 3 km AMSL, in agreement with the Spanish and British aviation authorities. The flight remained confined to Spanish and British airspaces.

As the ULA is not an aircraft, it cannot take off in strong wind or rainy conditions. It was therefore necessary to wait until
the weather conditions were suitable for take-off, i.e. with winds between 10 and 15 $ms^{-1}$ at ground level. The wind conditions also had to be associated with the presence of significant aerosol layers above the area of Gibraltar that was the subject of this airborne experiment. Between 1 and 22 August, when the ULA was available, only Saturday 13 August in the late afternoon (1950-2120 local time (LT)) met these conditions.



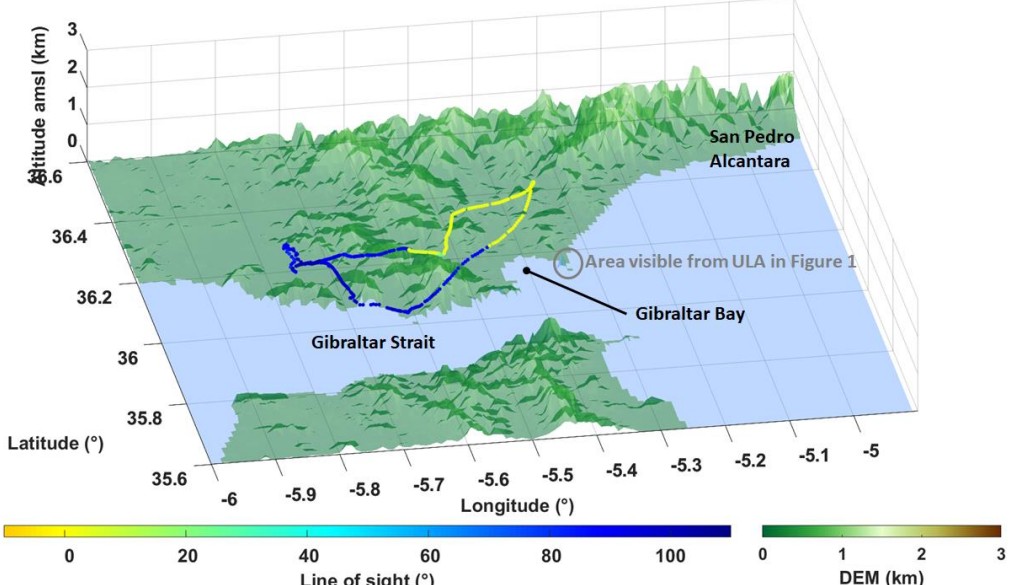

**Figure 2: Flight plan of the ULA above the Strait of Gibraltar on 13 August 2011. The colour bar represents the angle of the line of sight. The blue broken line corresponds to both the ascent and the descent and the yellow broken line to the ULA's flight ceiling. The view from the ULA in Figure 1 is also highlighted.**

## 2.2 Ground-based tools

The sea shore site of San Pedro Alcantara (see Figure 1, ~50 km Northeast of Gibraltar) was equipped with the first version of Lidar Automatic for Atmospheric Surveys using Raman Scattering (LAASURS, Chazette et al., 2017, 2019) described in Royer et al. (2011). LAASURS is an $N_2$-Raman lidar dedicated to research activities. It comprises three channels for the parallel and perpendicular polarizations with respect to the laser emission and the inelastic nitrogen vibrational Raman line of the laser induced atmospheric backscattered signal. The version used during this field campaign includes a Nd:YAG laser (Ultra® manufactured by Lumibird/Quantel) emitting 16 mJ at 355 nm collimated to fulfill eye-safety requirements. On the same site, a sunphotometer was installed and linked to the AErosol RObotic NETwork (AERONET) that distributes the data (https://aeronet.gsfc.nasa.gov/).

The ground-based site was operational from 25 June to 23 August 2011. Within the framework of this study, which focuses on the ULA flight, the interest is on the period from 11 to 14 August where significant aerosol optical thicknesses (AOT) were observed, as shown in Figure 3. These AOTs may correspond to a mix of different aerosol types along the time period when considering the Ångström exponent range (~0.3 to 1.3) also given in Figure 3.



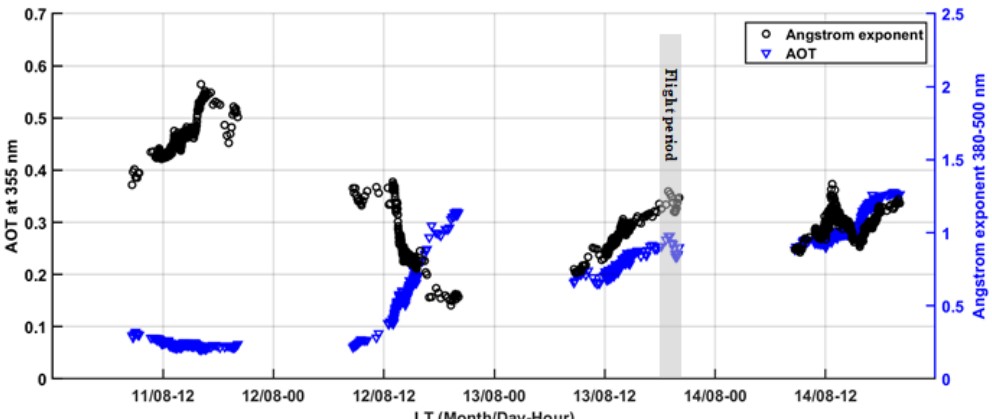

**Figure 3:** Temporal evolution in local time (LT) of both the aerosol optical thickness (AOT) and Ångström exponent. The flight period is highlighted by the gray area.

### 3 Data processing for lidar measurements

Whether for airborne observations (horizontal or nadir sightings) or from ground level, after the background correction, the range corrected lidar signal, also called the apparent backscatter coefficient (ABC), $S$ is written for the distance $s$ from the emitter as

$$S(s) = C \cdot \beta_\pi(s) \cdot F(s) \cdot exp\left[-\int_0^s \left(\left(1 + \left(\frac{\lambda_R}{\lambda_E}\right)^{-4}\right) \cdot \alpha_m(s') + \left(1 + \left(\frac{\lambda_R}{\lambda_E}\right)^{-A(s')}\right) \cdot LR(s') \cdot \beta_{\pi a}(s')\right) \cdot ds'\right]$$

Where $C$ is the system constant and $F$ is the overlap function. The backscatter coefficient $\beta_\pi$ is the sum of the molecular

and aerosol ($\beta_{\pi a}$) backscatter coefficients for the emitted wavelength $\lambda_E$ or the Raman backscatter coefficient for the $N_2$-Raman wavelength $\lambda_R$. The molecular extinction $\alpha_m$ is given at the emitted wavelength $\lambda_E$. It is worth noting that for the airborne lidar and the elastic channel of the ground-based lidar $\lambda_R = \lambda_E$, whereas for the $N_2$-Channel of LAASURS the wavelengths are different and it is necessary to take into account the Ångström exponent $A$ derived from the sunphotometer. The extinction to backscatter ratio (lidar ratio, LR) is given at the wavelength $\lambda_E$ and it can depend on

the altitude.

### 3.1 Airborne lidar

The inversion of airborne lidar data is performed in two successive steps, using the flight plan configuration that mixes horizontal and nadir sights. This approach was first used by Chazette et al. (2007), where it is discussed. As a reminder, the aerosol extinction coefficient $\alpha_a$ (AEC) is calculated using horizontal sighting measurements (here at ±10°), assuming

that the aerosol optical properties do not change along the line of sight between two distances $s$. During this flight, the lowest variability is obtained for $s$ between 0.7 and 1.2 km, which allows getting as far as possible from a residual effect of the overlap factor. Hence, for a flight altitude $z_f$, the AEC is given by:

$$\alpha_a(z_f) = \frac{1}{2}\frac{\partial Ln\big(S(s, z_f)\big)}{\partial s} - \alpha_m(z_f)$$

An estimate of the LR can then be calculated by adjusting the inversion (Klett, 1981) of the nadir-looking lidar profiles

to the AEC profile. There are several independent measurements to differentiate two aerosol layers against the altitude and to evaluate their respective LR. The separation altitude $z_0$ between the two layers is also evaluated and the continuity of the LR is ensured via a sigmoid function defined against the altitude $z$ expressed in km as





$$1/_{LR(z)} = 1/_{LR_l} + \left(1/_{LR_u} - 1/_{LR_l}\right) / \left(1 + e^{(z-z_0)/2.5}\right)$$

$LR_l$ and $LR_u$ correspond to the LR for the lower and upper aerosol layers, respectively.

The ULICE system also allows to evaluate the volume depolarisation ratio (VDR) and then the linear particle depolarisation ratio (PDR), as presented in Chazette et al. (2012).

Figure 4 shows the vertical profiles in nadir sighting for the ABC and VDR. They are obtained in the middle of the Strait of Gibraltar. They show the presence of the two aerosol layers between the sea surface and $z_0 \sim 1$ km AMSL, a transition to a less scattering and also low depolarizing layer, and then a second layer at altitude from 2 km AMSL. The ULA has flown within the last aerosol layer and we do not see its vertical extension that stretches beyond 3 km AMSL. This layer appears more marked because the associated backscattered signal is less attenuated, being close to the flight altitude. The

red layer in Figure 4, close to the sea level (~200 m AMSL), correspond to the marine boundary layer (MBL). The VDR is lower due to spherical aerosol presence.

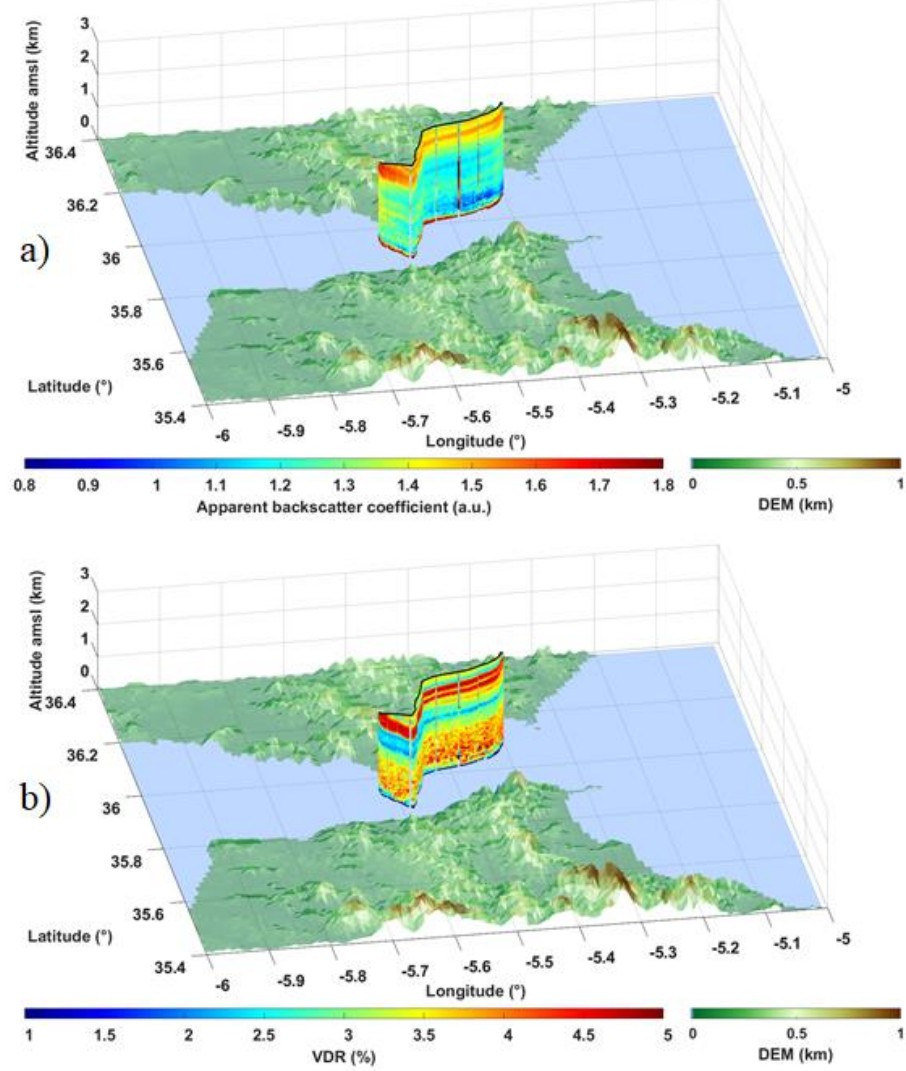

**Figure 4: Lidar profiles derived from ULICE onboard the ULA above the Strait of Gibraltar: a) the apparent backscatter coefficient and b) the volume depolarization ratio (VDR).**

**3.2**    **Ground-based lidar**

The retrieval of aerosol optical properties from the ground-based lidar is based on the approach previously presented in various papers and used both the synergy with the sunphotometer and the coupling between the elastic and $N_2$-Raman channels of LAASURS (e.g. Chazette et al., 2016; Royer et al., 2011). The signal to noise ratio (SNR) of the daytime lidar profiles is insufficient to use the $N_2$-Raman channel to reach the upper aerosol layers above 3 km AMSL and makes it difficult to identify different LR values. For this study, the inversions are performed with 30-minute time averages in order to evaluate an equivalent LR value between 0.4 (excluding the residual effect of the overlap factor) and 3 km AMSL. During the day, photometric data are also used, assuming a constant extinction value between the ground and 0.4 km AMSL. For special cases, a "two-layers" type inversion is performed as in Dieudonné et al. (2017) in order to verify that an equivalent LR does not induce significant uncertainties on the other optical parameters. As for airborne measurements, the PDR is also calculated for each lidar profile following Chazette et al. (2012).

The main uncertainties sources for this ground-based lidar are discussed in Royer et al. (2011). The uncertainty in the determination of the equivalent LR is in the range of 10-15 sr. The SNR limits the exploitable range of the lidar profile, as shown in Table 2 of Dieudonné et al. (2017). The relative uncertainties on the PDR are close to 10% for the AOTs encountered at 355 nm (AOT> 0.2, Figure 3).

### 3.3 Cross-calibration

Apart from the overlap factor, which is determined based on horizontal sighting, it is necessary to check the calibration of the parallel and perpendicular (via the VDR) channels between the two lidars so that the derived products can be compared. VDR calibrations are carried out independently of each other, according to the procedure presented in Chazette et al. (2012). They should result in identical vertical profiles for both instruments. The cross-comparison experiment was performed at the San Pedro Alcantara site on 6 August 2011, a few days before the flight. The results are shown in Figure 5. The different vertical structures are coherent. The ABC profiles (Figure 5a) match very well with a good agreement considering the molecular signal above the aerosol layer (> 4 km AMSL). The absolute deviation on the VDR is less than 0.2% (Figure 5b), leading to an absolute error of less than 2% on the PDR for the aerosol layers encountered during the experiment.

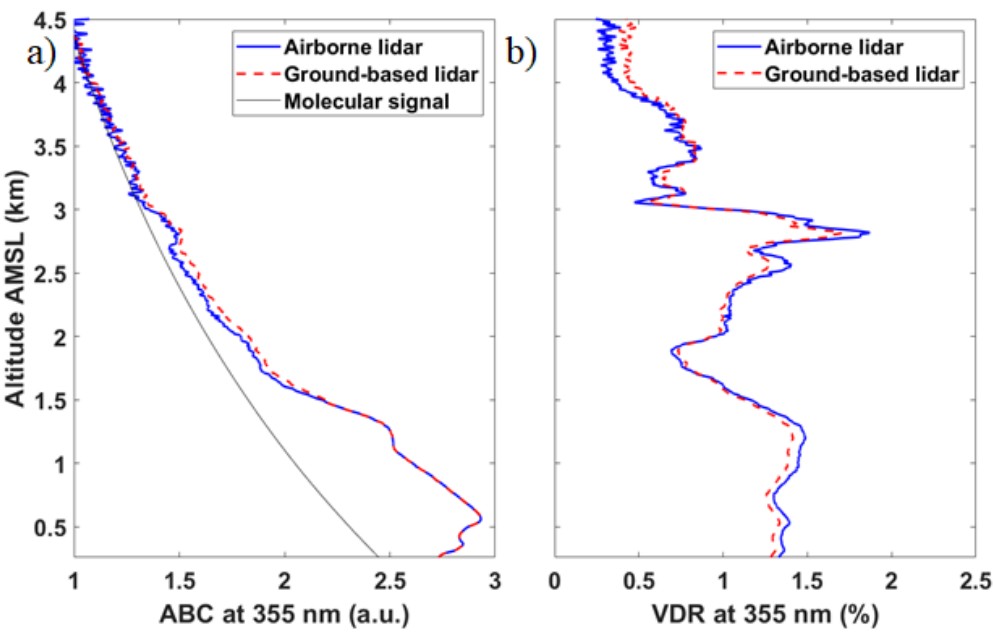





**Figure 5: Vertical profiles of the a) apparent backscatter coefficient (ABC) and b) volume depolarization ratio (VDR) derived from the ground-based lidar LAASURS and the airborne lidar ULICE.**

## 4 Vertical profiles of aerosol optical properties

The lidar observations will be analyzed in two stages. In the first stage, the aerosol layers over the Strait of Gibraltar and their associated optical properties will be studied. The link with ground-based lidar measurements at San Pedro Alcantara will be discussed in a second stage.

### 4.1 Aerosol optical properties from the airborne lidar

In Figure 6a, the vertical profile of the AEC retrieved from horizontal sighting is shown for the flight between 1950 and 2120 LT on 13 August 2011. A very close profile has been derived using nadir sighting when considering a LR ~ 26±2 sr below 1 km AMSL and a LR ~ 45±10 sr above this altitude. The first corresponds to what is expected for marine aerosols (e.g. Chazette et al., 2019; Flamant et al., 2000) while the second corresponds more to Saharan dust aerosols (e.g. Papayannis et al., 2008; Soupiona et al., 2018). Using both the LR and AEC, Figure 6b shows the corresponding vertical profile of the PDR. The two profiles match very well with a lower PDR value (~5%) within the MBL, as expected. The MBL presents a higher variability of the AEC, which may not only be linked to the heterogeneity of the wind field in the Strait of Gibraltar, but also to a strong maritime activity with the presence of numerous tankers. The PDR at 355 nm is between 10 and 15% within the dust layer, which appears low for a potential layer of Saharan aerosols for which one would expect values between 20 and 30% (e.g. Freudenthaler et al., 2009). Note that lower values have been reported by Papayannis et al. (2008) or Chazette et al. (2016) (~10-27%), and Soupiona et al. (2019) (11±1-34±2%).

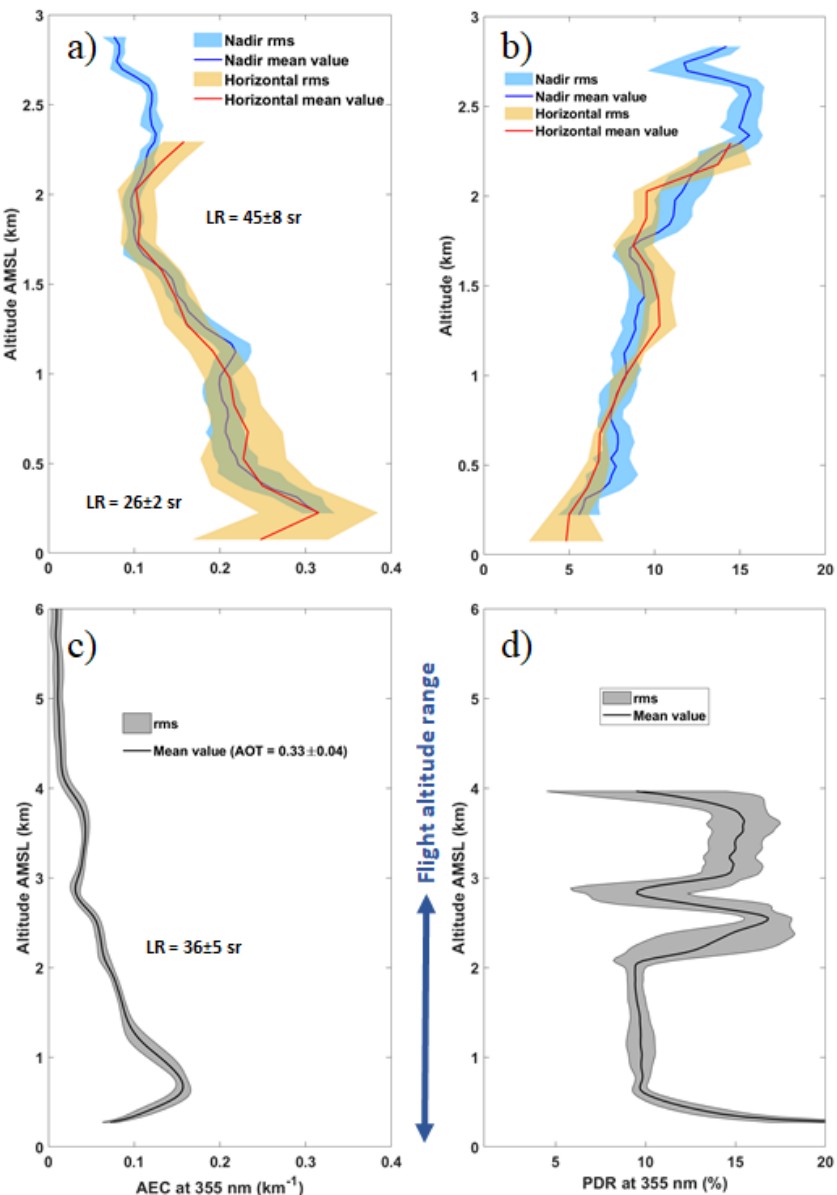

**Figure 6: Vertical profiles of a) the aerosol extinction coefficient (AEC) derived from the airborne lidar for horizontal and nadir sightings, b) the particle depolarization ratio (PDR) derived from the airborne lidar for horizontal and nadir sightings, c) the AEC derived from the ground-based lidar and d) the PDR derived from the ground-based lidar. The ground-based profiles correspond to an average between 1900 and 2200 local time on 13 August 2011. The aerosol optical thickness (AOT) is also given in Figure 6c.**

### 4.2   Link with the ground-based lidar

Only ground-based lidar observations between 11 and 15 August will be considered here. This period allows framing the airborne measurements by showing the atmospheric conditions before and after the flight. Figure 7 shows the temporal evolution of the AEC and PDR profiles, the equivalent LR in the aerosol column and the AOT. A first event can be identified on 11 August with AOTs~0.5 at 355 nm. It appears to diminish on 12 August to start again on 13 August. The





presence of depolarizing particles is shown from the PDR to ~6 km AMSL on 11 and 13 August. It is this presence, with a favourable weather forecast, that triggered the flight of 13 August in the late afternoon, with an AOT~0.35 at 355 nm. The mean AEC and PDR profiles derived from the ground-based lidar during the flight period are shown in Figure 6c and Figure 6d, respectively. Even if the shape of the AEC profile is similar to that retrieved from the airborne

measurements between 0 and 3 km AMSL, the amplitude is lower by a factor of more than 2. The PDR nevertheless appears consistent between the airborne and ground-based measurements in the dust-like layers, with values close to 15% in the layer between 2 and 3 km AMSL. Near the surface, high PDR values (>15%) are observed over San Pedro Alcantara. These values may be associated with local uprisings of dust aerosols (reported by visual observations at the site), less present before the afternoon of 13 August (Figure 7b). On the late afternoon of 13 August, the LR value is

intermediate (36±5 sr) to those of the two layers identified from airborne measurements (from 26±2 to 45±8 sr). Figure 8 shows the inversion of the average profile in Figure 6c using a two-layer distribution of the LR similar to that considered for vertical profiles from ULICE. The adjustment leads to similar LR values for the upper layer (34±4 sr) compared to an inversion with a constant LR. The discrepancy with the vertical profiles retrieved above the Strait of Gibraltar is mainly in the lower layer, where the value of 45±6 sr is more in favour of the presence of dust-like aerosols than marine aerosols

for San Pedro Alcantara. Nevertheless, we note the presence of aerosols of marine origin below 1 km AMSL before noon on 13 August. During the day on 12August, low layer structures are observed in Figure 7b, which can suggest mixtures between marine and dust aerosols. The values of the LR can then go below 35 sr, and can drop to ~25 sr.

Since the two lidars are consistent when measuring at the same site, the observed differences are therefore related to different local emissions for the lower layers and different transport processes for the upper layers, although the distance

between the two measurement points is only about 50 km. Nevertheless, it is worth noting that the range of LR values found in the literature for dusts is quite wide, ranging from 28 sr (Soupiona et al., 2019) to 80 sr (Papayannis et al., 2008). Intermediate values are reported in the Aegean Sea by Giannakaki et al. (2010) (52±18 sr) and Siomos et al. (2018) (Spring ~47±13 sr, Summer ~60±17 sr, Autumn ~47±15 sr). Over the Iberian Peninsula, Fernández et al. (2019) report LR between 44 and 55 sr corresponding to an extreme Saharan dust event intrusion. Such variability may make the LR

indiscriminate in the identification of atmospheric aerosols.





**Figure 7: Ground-based lidar-derived temporal evolution of the vertical profiles of a) the aerosol extinction coefficient (AEC), the particle depolarization ratio (PDR) and c) the lidar ratio (LR). In Figure 7c, the aerosol optical thickness (AOT) as derived from both the ground-based lidar and sun photometer, are also shown. The time location of the ULA flight over the Strait of Gibraltar is highlighted by the light-red area, around 2030 LT on 13 August 2011.**

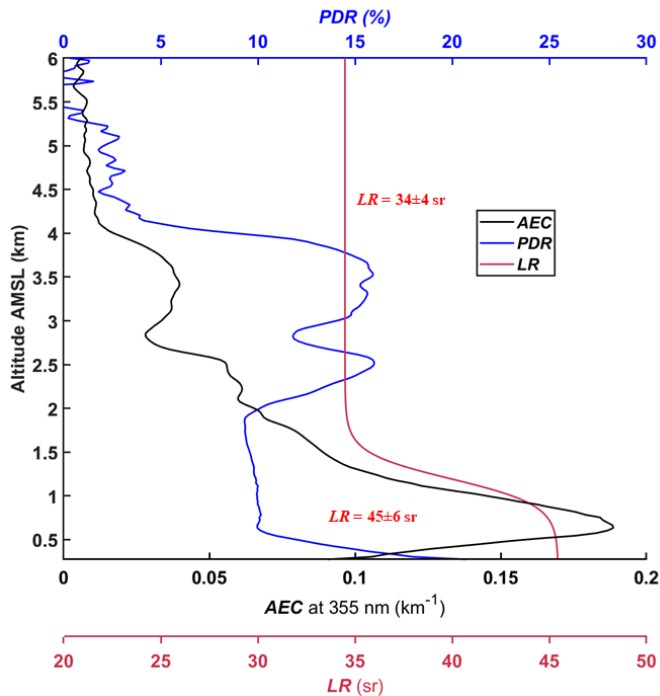

**Figure 8: Mean vertical profiles derived from the ground-based lidar between 1900 and 2200 LT on 13 August 2011: aerosol extinction coefficient (AEC), particle depolarization ratio (PDR) and lidar ratio (LR).**

## 5 Origin of aerosols observed above Gibraltar

Desert dust from North Africa is one of the main sources of aerosols over the Gibraltar area. Their transport is linked to large-scale meteorological conditions. On the decadal time scale, it has already been shown that the North Atlantic Oscillation (NAO) index can play a significant role in the occurrences of desert dust transport over the western Mediterranean (Moulin et al., 1997), with higher mean optical thicknesses during periods of positive NOA. The NAO index was negative over the period of the experiment, as shown in Figure 9 plotted from data recorded at the site

https://climatedataguide.ucar.edu/climate-data/hurrell-north-atlantic-oscillation-nao-index-station-based (Hurrell et al., 2004; Hurrell and Deser, 2010). Nonetheless, it is still somewhat higher than those encountered during most summer situations in the decade surrounding the period of the experiment. This intermediate regime therefore does not provide a clear view of the specificity of summer 2011 in terms of the occurrence of dust transport events over southern Spain.

From a local perspective, the particles influencing mainly the planetary boundary layer (PBL) close to the Gibraltar area

are related to marine emissions, but also to industrial activities, as shown by Gallero et al. (2006). The region of Gibraltar, near Algeciras, is indeed significantly industrialised with a refinery, a petrochemical factory, a steel factory, a coal power plant, a heavy fuel oil power plant and a paper factory, all of which emit particles and aerosol precursors.





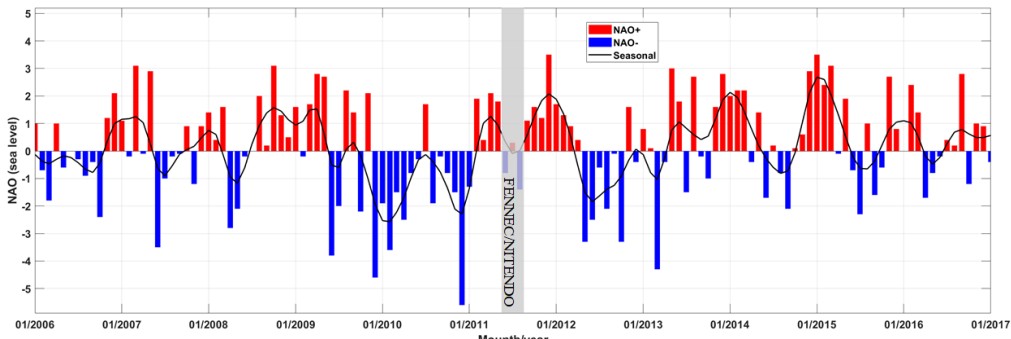

**Figure 9: Temporal evolution of the monthly average North Atlantic Oscillation (NAO) index. The data are those of the site https://climatedataguide.ucar.edu/climate-data/hurrell-north-atlantic-oscillation-nao-index-station-based. The grey area highlights the summer 2011 where the NITENDO field campaign took place.**

5   **5.1   Meteorological situation**

During the period of the airborne experiment, the Azores high was moving strongly northward (Figure 10) leading to a blockage of the flow from the African coast and even the establishment of a "north-south" circulation at 850 hPa over the Strait of Gibraltar. On 13 August 2011 (Figure 10a), the air masses arriving over the Strait of Gibraltar were coming from the north-west, passing over the Iberian Peninsula. This type of circulation is obviously not favourable for the transport
10   of dust from Saharan origin. It was different on 11 August, where a small depression was present west of the Strait of Gibraltar, facilitating the transport of aerosols from Morocco (Figure 10b). These two contrasting weather patterns may explain the evolution of the LR in the upper aerosol layer between 11 and 13 August.




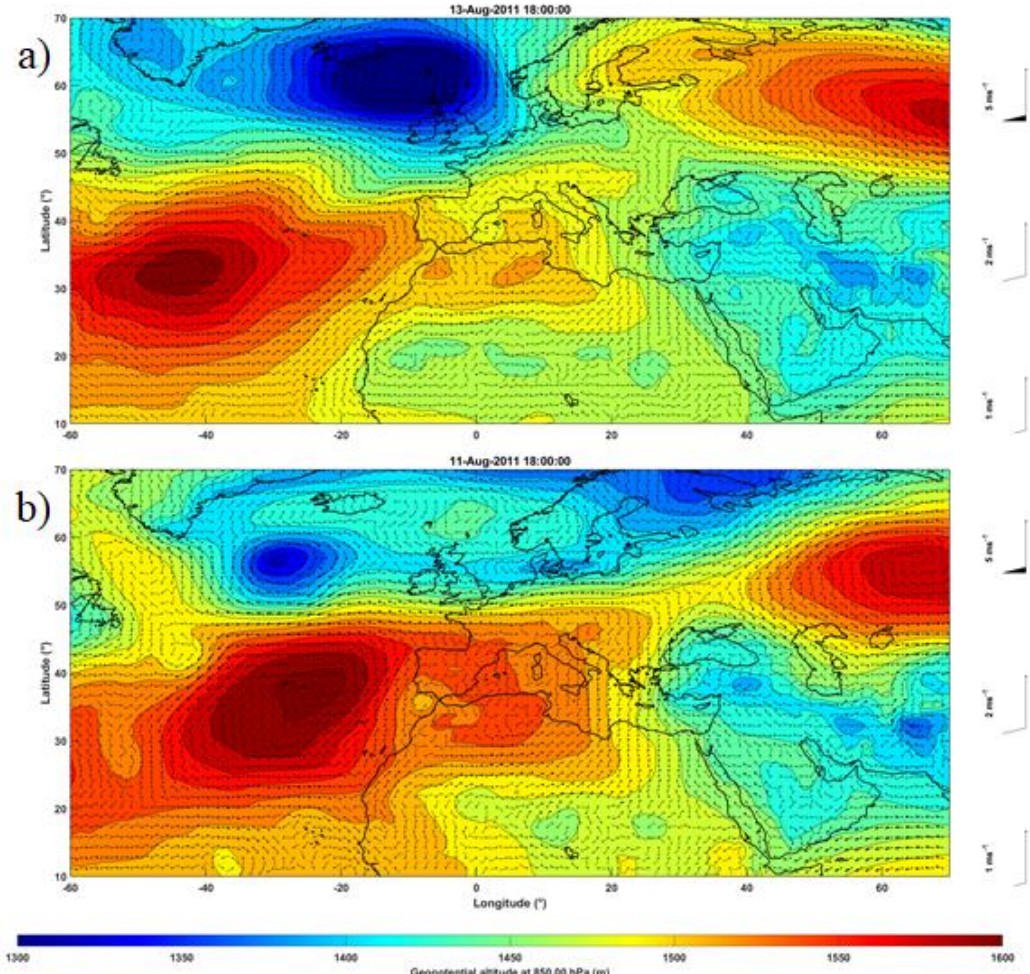

**Figure 10: Geopotential altitude at 850 hPa on a) 13 August 2011 at 1800 UTC and b) 11 August 2011 at 1800 UTC. The wind fields at the same pressure level are superimposed. The ERA5 reanalysis (https://www.ecmwf.int/ en/forecasts/datasets/archive-datasets/reanalysis-datasets/era5) with a horizontal resolution of 0.25° are used.**

## 5.2 Spaceborne observations

The location of aerosol plumes can be highlighted by the Moderate Resolution Imaging Spectroradiometer (MODIS) onboard the polar-orbiting platforms Terra and Aqua (King et al., 1992; Salmonson et al., 1989). The level 2 products are provided with a spatial horizontal resolution of $10 \times 10$ km$^2$ (http://modis.gsfc.nasa.gov).The uncertainty on the AOT is $\pm 0.15 \pm 0.05 AOT$ over land and $\pm 0.05 \pm 0.03 AOT$ over ocean (Chu et al., 2002). A combination of the aerosol optical thickness (AOT) at 550 nm derived from the two satellites is given in Figure 11. On 13 August 2011 (Figure 11a), a significant contrast is observed between the west and east of the Strait of Gibraltar, with a higher AOT (~0.4) at 550 nm to the west. This is consistent with what was inferred from the airborne lidar observations compared to those made on from the ground-based lidar over San Pedro Alcantara. On the contrary, on 11 August, similar AOTs (>0.6) are observed over the Atlantic Ocean and the Mediterranean Sea (Figure 11b). The higher AOTs are consistent with those retrieved from the ground-based lidar. The continuity of the aerosol plume between the Moroccan coast and the Strait of Gibraltar is more pronounced than on 13 August.



Complementary with the data from the MODIS, the vertical profiles of the aerosol layers are derived from the Cloud-Aerosol LIdar with Orthogonal Polarization (CALIOP) aboard Cloud-Aerosol Lidar and Infrared Pathfinder Satellite Observations (CALIPSO, http://www-calipso.larc.nasa.gov, Winker et al. (2007)). The 4.10 version of CALIOP level-2 data is used, whose aerosol typing have been i*m*proved (Burton et al., 2015). A night orbit (~0240 UTC) over the Atlantic

5 and a day orbit (~1240 UTC) just over the Strait of Gibraltar were used for August 13. They both show aerosol layers up to ~4 km AMSL for the studied area (Figure 12a). These layers are mainly identified as dusts and polluted dusts (Figure 12b). Above Gibraltar (daytime orbit), polluted dusts are predominant. They are also preponderant over the Iberian Peninsula, with even elevated smoke over land. This is therefore consistent with what has been found as a vertical structure and LR via the airborne lidar. At this stage, the upper aerosol layer that was sampled by the airborne lidar does not appear

10 to be pure dusts, but a mix that can be associated with air masses of different origins.

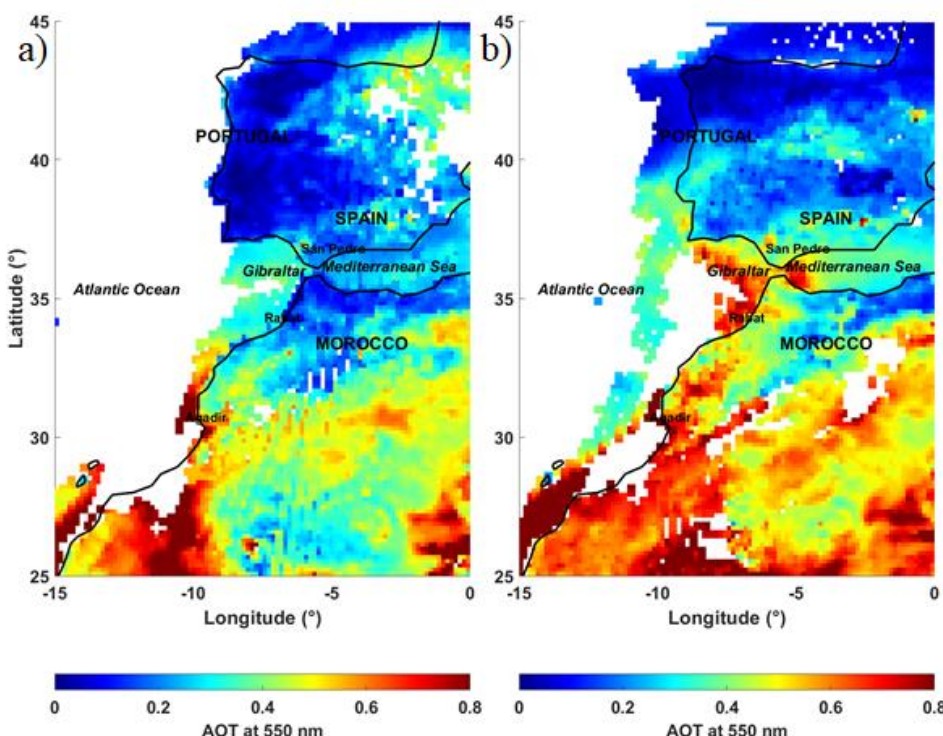

**Figure 11: Aerosol optical thickness (AOT) at 550 nm derived from MODIS on a) 13 August 2011 and b) 11 August 2011.**

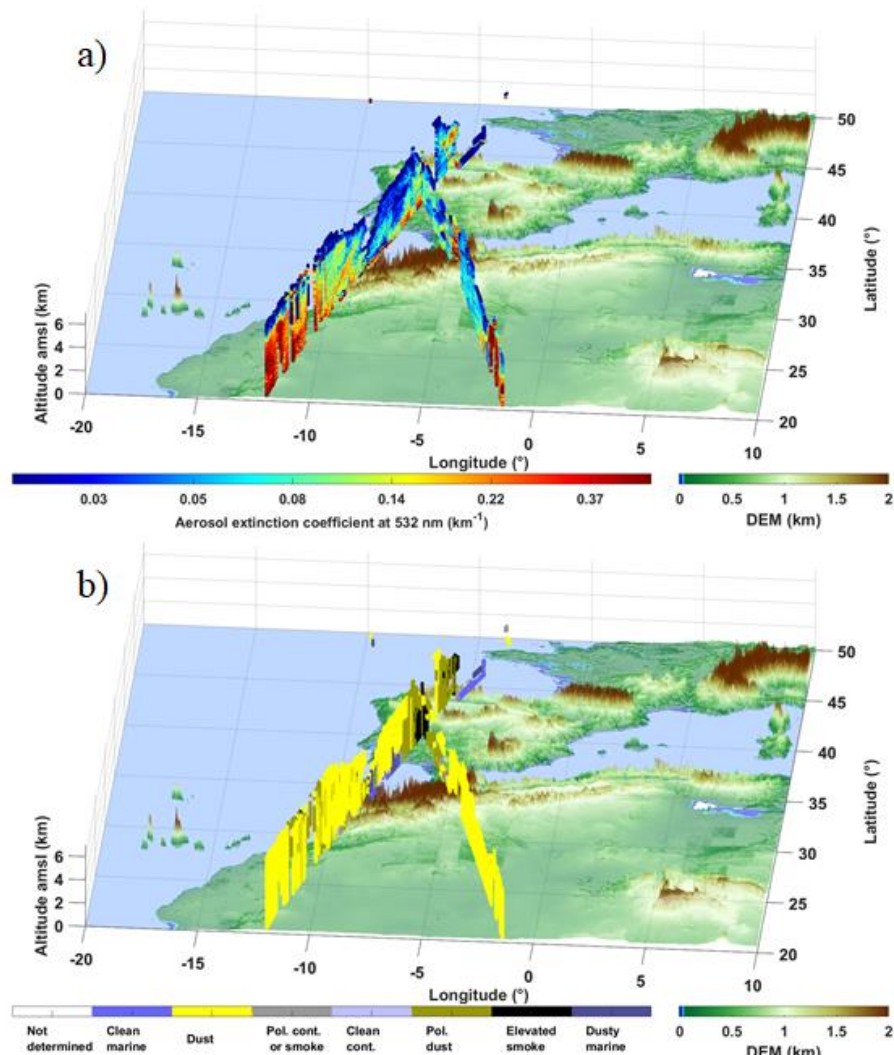

**Figure 12: CALIOP-derived a) aerosol extinction coefficient at 532 nm and b) aerosol typing. Two orbits are plotted on 13 August 2011, the first one during night time at ~0240 UTC (westernmost over North Africa) and the second one during day time at ~1340 UTC (easternmost over North Africa).**

## 5.3 Air mass origins

In order to better identify the origin of the aerosols observed during the field campaign, back trajectory studies were performed. For this, the Single Particle Lagrangian Integrated Trajectory (HYSPLIT, Stein et al. (2015)) was used. It was initialised according to lidar observations over the Strait of Gibraltar and San Pedro Alcantara. The wind fields used were taken from the Global Data Assimilation System (GDAS, http://www.ncep.noaa.gov/) at the horizontal resolution of 0.5°. HYSPLIT operated in ensemble mode, i.e. 27 back trajectories are computed for each end location and for different altitude ranges with a vertical sampling of 250 m.

Most of the aerosol layers observed by the airborne lidar and daytime orbit of CALIOP are located below 2.5 km AMSL. As shown in Figure 13a, the air mass comes very clearly from the Iberian Peninsula for these altitudes. The aerosol type is reminiscent of local uptakes of terrigenous dusts that may be mixed with pollution aerosols. Above 2.5 km AMSL, the air mass comes from the tropical Atlantic and may have trapped Saharan dust aerosols. For the night CALIOP orbit on



13 August (~0240 UTC), although the trajectories are significantly different, the origin is also the Iberian Peninsula (Figure 13b), and the same type of aerosol is likely to be observed over the Atlantic off Gibraltar until 4 km AMSL. Further north, a contribution from forest fires cannot be excluded, but no satellite observations clearly identify them. Above San Pedro Alcantara, at 2100 UTC (Figure 13c), similar trajectories are observed as the ones over Gibraltar. The

LR is nevertheless quite different because its calculation integrates layers not accessible to the airborne experiment, above 2.5 km AMSL. It should be noted that the back trajectories below 500 m AMSL are not taken into consideration as they are not significant with regard to the topography of the Spanish coast. The back trajectories in Figure 13d are calculated on 11 August and show that the differences in LR and PDR observed between 13 and 11 August 2011 over San Pedro Alcantara are explained by very different origins of the air masses. On 11 August, the probable source of the aerosols is

located in Morocco. This conclusion can be supported using the brightness temperature anomaly (BTA) calculated over the month of August 2011 from the 10.8 μm channel of the Spinning Enhanced Visible and InfraRed Imager (SEVIRI, Schmetz et al. (2002)) following an approach similar to that proposed by Legrand et al. (1992). The BTA on 11 August is given in Figure 14. It reveals very clearly the presence of dust aerosols northeast of Agadir in the way of the back trajectories plotted in Figure 13d. It should be noted that no active source is detectable by the same approach on 13 August

2011.









**Figure 13: Bidimensional histogram derived from back trajectories computed using HYSPLIT on a) 13 August 2100 UTC over the flight location, b) 13 August 0300 UTC at the central position of CALIOP ground track off Gibraltar, c) 13 August 2100 UTC over San Pedro Alcantara and d) 12 August 0000 UTC over San Pedro Alcantara. The aerosol transport altitudes are indicated for each main trajectory.**

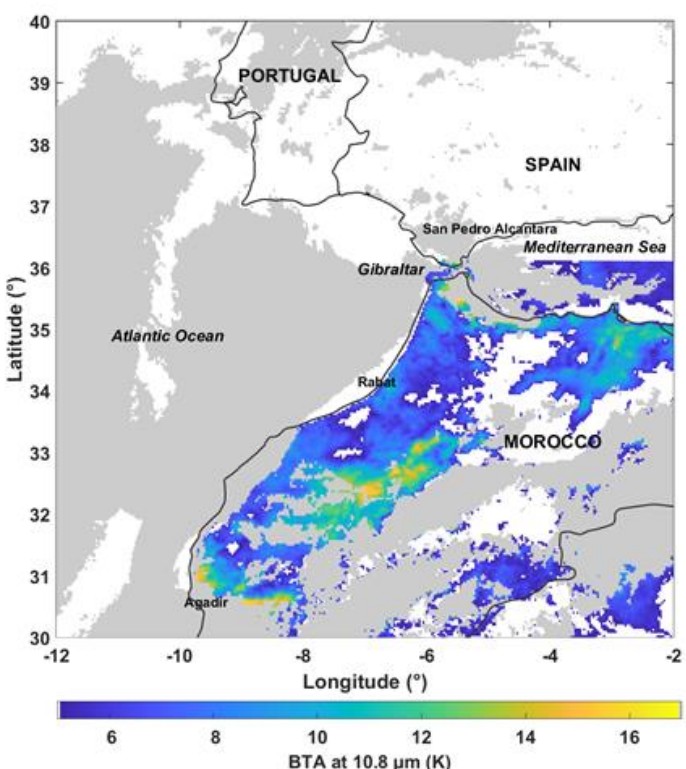

**Figure 14: Brightness temperature anomaly (BTA) on 11 August 2011 at 1200 UTC. The cloud mask has been applied in grey.**

## 6    Conclusion

The western end of the Mediterranean, which connects with the Atlantic Ocean, is one of the areas most subject to the transport of terrigenous aerosols. This can be explained by the passage of lows, which, together with the Azores high,

favours the transport of desert aerosols through the Strait of Gibraltar towards the Iberian Peninsula and along an inverse path towards northwest Africa. Nevertheless, there have been few studies on the characterisation of the vertical distribution of aerosols over this region. Although punctual in time, like many other studies, the SUSIE experiment has provided useful information on vertical profiles of aerosol optical properties. The profiles were obtained from airborne lidar observations over Gibraltar and ground-based $N_2$-Raman lidar measurements nearby Gibraltar, at San Pedro

Alcantara, located ~50 km east of Gibraltar. Over the Strait of Gibraltar, the aerosol extinction coefficient (AEC), particle depolarization ratio (PDR) and lidar ratio (LR) were derived from a flight plan that allowed separate retrievals of the aerosol extinction and backscatter properties. Although the distance between the two measurement sites is small, the optical properties of the aerosols proved to be significantly different. The planetary boundary layer was more influenced by terrigenous aerosols over San Pedro Alcantara, whereas the marine aerosols were dominant over Gibraltar. In the lower

free troposphere, the difference between the LRs, 45±8 sr for Gibraltar and 34±4 sr for San Pedro Alcantara, is somewhat less noticeable and can be attributed to a higher weighting of the upper atmospheric layers as sampled by the ground-



based lidar. The back trajectories show a dichotomy between the air masses below and above 2.5 km AMSL and thus a possible mix between continental and Saharan aerosols, respectively. This would lead to think that the continental terrigenous aerosols would have a LR of about 45 sr while the Saharan aerosols would have a LR of less than 34 sr. This is also what is found in Figure 7 during the Saharan aerosol event of Moroccan origin on 11 August 2011. A mixture of

varying proportions of these different types of terrigenous particles is likely, to which are added pollution or biomass burning aerosols in varying quantities. All this can explain the range of variation of the LR at 355 nm that is deduced from the scientific literature.

The use of LR look-up tables for the inversion of satellite lidar measurements can therefore lead to biased results in situations such as those encountered during SUSIE. Using the CALIOP classification in the context of this work, polluted

dusts should be classified with LR ~45 sr and Saharan dusts with LR less than 34 sr for the wavelength of 355 nm. Hence, the LR variation is inversed compared with what is considered for CALIOP (44 sr for dusts and 55 sr for polluted dust at 532 nm) and even for  the Cloud-Aerosol Transport System (CATS, Yorks et al., 2016) where LR = 45 sr for dusts and LR = 35 sr for polluted dusts at 532 nm. It therefore appears important to update its classifications in the perspective of the analysis of lidar profiles from the ADM-AEOLUS mission, and also from the future EARTHCARE mission.

**Acknowledgments.** This work was supported by the Centre National d'Études Spatiales (CNES) and by the Commissariat à l'Energie Atomique et aux Énergies Alternatives (CEA). Joseph Sanak is acknowledged for his help during the field experiment. The ULA flights were performed by Santi Font. The author would like to thank the AERONET network for sunphotometer products (at https://aeronet.gsfc.nasa.gov/), the MODIS Science, Processing and

Data Support Teams for producing and providing MODIS data (at https://modis.gsfc.nasa.gov/data/dataprod/), and the NASA Langley Research Center Atmospheric Sciences Data Center for the data processing and distribution of CALIPSO products (level 4.10, at https://eosweb.larc.nasa.gov/HORDERBIN/HTML_Start.cgi). The NOAA Air Resources Laboratory (ARL) is acknowledged for the provision of the HYSPLIT transport and dispersion model and READY website (http://www.ready.noaa.gov) used in this publication. The ERA5 dataset is provided by the European Centre for

Medium-Range Weather Forecasts integrated forecast system (ECMWF), developed through the Copernicus Climate Change Service (https://climate.copernicus.eu/).

**Competing interests.** The authors declare that they have no conflict of interest.



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
