# Peer review of "Aerosol optical properties as observed from an ultralight aircraft over the Strait of Gibraltar"

_Atmospheric Measurement Techniques, 2020_

## Referee Comment (RC1) · Anonymous Referee #2 · 10 May 2020

The manuscript analyzes the properties of aerosol over Gibraltar, derived from observations of elastic backscatter lidar, installed on an ultra-light aircraft, and from the coastal site Raman lidar. The manuscript is clearly and well written and can be published after some minor revisions. 1. Equations should be numbered. 2. P.5 ln8. The equation is formally correct, but reader can be confused, because author first writes "N2-Raman wavelength ðÌlJĘðÍŚĚ" and later "the ground-based lidar ðÌlJĘðÍŚĚ=ðÌlJĘðÍŘÿ,". The reference for original work of Ansmann et al., 1992 should be provided. 3. P.10 ln 21. "the range of LR values found in the literature for dusts is quite wide, ranging from 28 sr (Soupiona et al., 2019) to 80 sr (Papayannis et al., 2008)." Lidar ratio of dust at 355 nm strongly depends on the imaginary part (Im) of the refractive index. (e.g. https://doi.org/10.5194/acp-2020-98). So low values of LR

observed in this work may indicate to low Im in UV. 4. P.20 ln 3. "Saharan aerosols would have a LR of less than 34 sr." I think this LR is too low for dust. It can be due to mixing with with maritime particles. 5. Conclusion. P.20 ln 9. "Using the CALIOP classification in the context of this work, polluted dusts should be classified with LR ∼45 sr and Saharan dusts with LR less than 34 sr for the wavelength of 355 nm." I think this statement is unsupported. There are numerous Raman lidar measurements in Africa (e.g. SAMUM, SHADOW experiments), bringing LR at 355 nm well above 40 sr.

---

## Referee Comment (RC2) · Anonymous Referee #1 · 20 May 2020

General comments This paper reports observations using a lidar on an ultralight aircraft and a ground-based Raman lidar. The subject is interesting, and the manuscript is generally well written. However, it requires minor revisions.

Specific comments Page 4, line 7: "(See Figure 1, .." should be "(See Figure 2, .."

Figure 2: What is the definition of the angle of the line of sight? (What is the angle mentioned in line 20 of page 5 "+- 10deg"? What is the definition?).

Figure 4: It would be more impressive if the vertical distributions along the aircraft path are discussed in more details.

Page 7. 3.3 Cross-calibration: The legend of Figure 5 "Airborne lidar" is confusing. It is not airborne. It should be clearly described that both lidars are pointing vertically from

the ground.

Figure 7 and the figure caption: "NITENDO"? "SUSIE" isn't it?

---

## Author Comment (AC1) · 30 Jun 2020

*Aerosol optical properties as observed from an ultralight aircraft over the Strait of Gibraltar*

*by Patrick Chazette*

**Response to the referee #2 comments**

The manuscript analyzes the properties of aerosol over Gibraltar, derived from observations of elastic backscatter lidar, installed on an ultra-light aircraft, and from the coastal site Raman lidar. The manuscript is clearly and well written and can be published after some minor revisions.

Equations should be numbered.

**Agree. The numbering has been added.**

2. P.5 ln8. The equation is formally correct, but reader can be confused, because author first writes "N2-Raman wavelength ðˊIIJ ¸ EðˊI ´SEˇ " and later "the ground-based lidar ðˊIIJ ¸ EðˊI ´S ˇ E=ðˊIIJ ¸ EðˊIRˇ ÿ,".

The reference for original work of Ansmann et al., 1992 should be provided.

**Agree. The correction has been done and the reference to Ansmann et al. (1992) has been added.**

3. P.10 ln 21. "the range of LR values found in the literature for dusts is quite wide, ranging from 28 sr (Soupiona et al., 2019) to 80 sr (Papayannis et al., 2008)." Lidar ratio of dust at 355 nm strongly depends on the imaginary part (Im) of the refractive index. (e.g. https://doi.org/10.5194/acp-2020-98). So low values of LR observed in this work may indicate to low Im in UV.

**This is an interesting piece of information that has been added in the text as: "Veselovskii et al. (2020) explained that Lidar ratio of dust aerosols at 355 nm above Senegal strongly depends on the imaginary part of the refractive index and that so low values of LR observed in this work may indicate to low imaginary part.".**

4. P.20 ln 3. "Saharan aerosols would have a LR of less than 34 sr." I think this LR is too low for dust. It can be due to mixing with with maritime particles.

**The available data do not allow us to understand why the LR is low. Airborne in situ measurements should be available, but even then, the uncertainties in such measurements are often too large to conclude. This part of the conclusion has been reviewed to draw the mixing hypothesis: " For such low values, there may be a mixture of marine particles in the upper aerosol layer. It may be generated by a recirculation at altitude above the PBL top of a certain quantity of marine aerosols above the coastal site (e.g. Chazette et al., 2019), but no strong argument is available to claim this, therefore that statement remains speculative. As a result, we can infer that a mixture of different types of particles is likely, to which pollution or biomass burning aerosols in varying quantities may be added. ".**

5. Conclusion. P.20 ln 9. "Using the CALIOP classification in the context of this work, polluted dusts should be classified with LR 45 sr and Saharan dusts with LR less than 34 sr for the wavelength of 355 nm." I think this statement is unsupported. There are numerous Raman lidar measurements in Africa (e.g. SAMUM, SHADOW experiments), bringing LR at 355 nm well above 40 sr.

**It is not as a general conclusion, but on the case studied in this article. That's why I added " in the context of this work ". To avoid misunderstandings, I've added the precision: " It is worth noting that there are numerous other Raman lidar measurements in Africa bringing LR at 355 nm well above 40 sr (e.g. SAMUM (Ansmann et al., 2011), SHADOW (Veselovskii et al., 2020) experiments)."**

---

## Author Comment (AC2) · 30 Jun 2020

*Aerosol optical properties as observed from an ultralight aircraft over the Strait of Gibraltar*

*by Patrick Chazette*

**Response to the referee #1 comments**

General comments This paper reports observations using a lidar on an ultralight aircraft and a ground-based Raman lidar. The subject is interesting, and the manuscript is generally well written. However, it requires minor revisions.

Specific comments Page 4, line 7: "(See Figure 1, .." should be "(See Figure 2, .."

**Agree. The correction has been done.**

Figure 2: What is the definition of the angle of the line of sight? (What is the angle mentioned in line 20 of page 5 "+- 10deg"? What is the definition?).

**This definition is now better explained in the text and given in the caption of Figure 2: " Flight plan of the ULA above the Strait of Gibraltar on 13 August 2011. The colour bar represents the angle of the line of sight (with respect to the true horizon).".**

Figure 4: It would be more impressive if the vertical distributions along the aircraft path are discussed in more details.

**Agree. An explanation of the evolution of the profiles along the aircraft path has been added: " Unlike the evolution of VDR profiles along the ULA path, the layered structure of the ABC evolves significantly. The upper aerosol layer is more intense in the western part than in the eastern part of the strait of Gibraltar. There also appears to be more aerosol around 1 km AMSL on the west side.".**

Page 7. 3.3 Cross-calibration: The legend of Figure 5 "Airborne lidar" is confusing. It is not airborne. It should be clearly described that both lidars are pointing vertically from the ground.

**Agree. The caption has been modified in this way using the name of the lidar.**

Figure 7 and the figure caption: "NITENDO"? "SUSIE" isn't it?

**It's a mistake, the correction has been made in both the caption and Figure 9.**